# Sibling Relationship Dynamics in Families with a Child Diagnosed with a Chronic Mental Disorder versus a Somatic Condition

**DOI:** 10.3390/children10030587

**Published:** 2023-03-19

**Authors:** Florina Rad, Emanuela Lucia Andrei, Alecsandra Irimie-Ana, Ilinca Olteanu, Magdalena Budișteanu, Ilinca Mihailescu, Elma-Maria Mînecan, Mihnea Costin Manea, Anca Coliță, Alexandra Buică

**Affiliations:** 1Psychiatry Hospital, 041914 Bucharest, Romania; florina.rad@umfcd.ro (F.R.); lucia.andrei@drd.umfcd.ro (E.L.A.); ilinca.mihailescu@umfcd.ro (I.M.); minecan.elma@yahoo.com (E.-M.M.); mihnea.manea@umfcd.ro (M.C.M.); anca.colita@umfcd.ro (A.C.); 2Child and Adolescent Psychiatry Department, “Carol Davila” University of Medicine and Pharmacy, 020021 Bucharest, Romania; alexandra.buica@umfcd.ro; 3Faculty of Sociology and Social Work, University of Bucharest, 050107 Bucharest, Romania; alecsandraana1@yahoo.com; 4Clinical Hospital for Infectious and Tropical Diseases, 030303 Bucharest, Romania; ilinca.olte@gmail.com; 5Fundeni Clinical Institute, Pediatric Department, 022328 Bucharest, Romania; 6Children Emergency Hospital, 011743 Bucharest, Romania

**Keywords:** chronic mental disorders, somatic conditions, siblings

## Abstract

Background: Recent research still focuses on the psychological impact on siblings and the problematic relationships in families with children with chronic illnesses. Our study evaluates the dynamics in sibling relationships in families with a child diagnosed with a chronic disease. Methods: We comparatively evaluated the degree of empathy, involvement, friendship, and rivalry in sibling relationships in two groups of families who have a child with a chronic pediatric disorder versus a chronic mental disorder. Results: The levels of involvement/friendship, empathy/care/concern, and education/learning were significantly higher in the pediatric group. Where there were siblings under the age of 10, rivalry scores tended to be higher in both groups. Conclusions: Coping strategies, emphatic interactions, and implications in common activities are difficult to identify in the relationship between siblings when one of them has a chronic mental disorder. All of these negative aspects entail poor quality sibling relationships and draw alarm signals regarding the need for monitoring and intervention familial programs.

## 1. Introduction

The first institution of socialization for an individual is the family and, at the family level, sibling relationships play an essential role in this complex process of human learning. Relationships between siblings can be based on cooperation, love, and mutual support; however, they can also contain rivalry, envy, and jealousy. In families where there is a child diagnosed with a chronic disease, it is expected that the relational dynamics will change [1]. Moreover, it is expected that a chronic mental disorder will produce different reactions compared with a chronic somatic condition, taking into account the stigma that characterizes the first category of disorders.

Several studies have found a link between the behavior of children in the peer group and the type of relationship they had with their siblings [2]. Other authors have observed that relationships between siblings also influence children’s abilities regarding emotional adjustment and integration, individually, For example, when there was a brother with domineering and controlling behavior, the chances of the other sibling developing a low self-esteem while also presenting externalization/internalization disorders, were very high [3].

Our analysis comparatively describes the relationship between siblings when one sibling in the family suffers from a chronic illness, be it somatic or psychiatric. In our exploratory analysis, we evaluated the competition between children for their caregiver’s attention and their position in the family structure. This rivalry was evaluated according to Piaget’s moral developmental stages [4], expecting that, in the healthy sibling, the perspective of competitiveness would change after the age of 10.

According to Piaget’s studies, the development of moral judgment goes through four stages. The third stage, which emerges around 10 years old, is characterized by a transformation at the level of perceiving the rules—rules are no longer perceived by the child as external, as imposed by adults, but as a result of their free judgment [5].

Most of the studies on this topic have focused on the influence of the family structure on the professional and emotional development of children, and less on how the pathology of one of the children influences the relationships dynamics. The available research focuses on overall family experiences, while our study brings new information related specifically to the siblings’ experiences. We considered the way these experiences influence the siblings’ relationships, and the possibility that rivalry may be augmented when one of the children has a chronic mental illness, as opposed to a somatic condition. In addition, most of the studies published so far evaluate the experience of siblings of children with chronic pediatric ailments [6,7], while this paper presents comparative observations depending on the chronic somatic or mental disorders.

The objectives of this study were as follows:Comparison of healthy siblings’ behaviors between two groups: families with children diagnosed with a mental illness and families with children diagnosed with a somatic condition.Comparison of the behavior of healthy siblings up to the age of 10 with that of those over 10 years of age.

## 2. Methods

Two groups of families were included in the study: a group consisting of 50 families with at least two children, in which there was a child diagnosed with a chronic mental disorder, and a group of 50 families, with at least two children, in which one of the children had been diagnosed with a chronic somatic condition. After signing the informed consent, one of the parents completed two evaluation instruments: a questionnaire regarding the demographic and family structure data and the Sibling Inventory of Behavior (SIB) [8]. This instrument evaluates sibling relationships in families with children with disabilities, without, however, differentiating between the nature of the disorder (mental/somatic) [9]. The instrument has more dimensions—evaluating the behavior between siblings, four of them targeting empathy/care for the disabled brother, involvement and leadership (guidance), and acceptance; the following scales evaluated anger, teasing, and the absence of an attitude of kindness, avoidance, and shame/embarrassment.

Only one parent gave his/her consent and filled out the questionnaires as well. The exclusion criteria were as follows: step siblings, children in foster care, siblings who did not live together, the presence of another adopted child in the family, or two siblings with disabilities. Only the behavior of the siblings aged 7–10 years and 11–18 years was evaluated.

The sampling was one of convenience; the subjects from the psychiatric group being selected from among the patients evaluated in the Child and Adolescent Psychiatry Clinic at “Prof. Dr. Al. Obregia” Psychiatry Hospital from Bucharest. The subjects from the somatic group were selected from pediatric services evaluating chronic diseases, belonging to the Fundeni Clinical Institute from Bucharest. The subjects included in the pediatric group (somatic disorders) were selected so that the brother whose behavior was evaluated by the questionnaire could be age-, sex, and origin-matched to the psychiatric group. Anonymization of the data was achieved by creating a register in which, after including the family in the study, a current number was recorded, a number that was found on the parents’ questionnaires. Connecting the data between the anonymized ID number and the original file was prevented by using different persons in different research phases, as well as the researchers involved in data management and statistical analysis not having access to the register or the informed consent.

The SIB questionnaire consisted of 32 Likert items (1–never, 2–rarely, 3–sometimes/sometimes, 4–frequently, 5–always), assessing six areas of interest: play/company, empathy/care, education/learning, rivalry, conflict/aggressiveness, and avoidance.

The statistical analysis was performed in the IBM SPSS v20 program. Descriptive tests, the Z test of proportions, Chi-Square, Likelihood Ratio, homogeneity tests, Mann−Whitney tests, and Factorial Anova were used to test inter-variable relationships.

The study data are not publicly available due to ethical concerns. Patient privacy and security are protected, according to the ethical rules of our institution and their restriction regarding data sharing. The study was conducted with the approval of the “Prof. Dr. Alexandru Obregia Psychiatry Hospital” Ethical Committee.

## 3. Results

In the somatic group, 46% of subjects came from the rural area and 54% from the urban area, while in the psychiatric group 58%, came from the rural area and the remaining 42% came from the urban area. Families in the somatic group had between 2 and 9 children, as follows: 74% had 2 children, 14% had 3 children, 4% had 3 children, 6% had 5 children, and 2% had 9 children. In the psychiatric group, the number of children were as follows: 64% had 2 children, 18% had 3 children, 14% had 4 children, 2% had 5 children, and 2% had 9 children. Z tests for comparing proportions showed that there were no statistically significant differences between the proportions of the number of children between the two groups (z2children = 1.08, *p* = 0.28; z3children = −0.54, *p* = 0.58; z4children = −1.75, *p* = 0.08; z5children= 1.02, *p* = 0.31). The proportion of families who had three female children was significantly higher in the psychiatric group compared with the somatic one (z = −2.46, *p* = 0.01).

In the somatic group, 1% of children had at least one unschooled parent, while 7% graduated from gymnasium level, 11% from high school, 6% from vocational school, 6% from post-secondary studies, and 19% from higher education (university/postgraduate). In the psychiatric group, 3% had a parent that never attended school, 9% graduated from the gymnasium level, 19% from high school, 5% from post-secondary studies, and 9% from higher education (university). The Chi-square test showed a significant difference between the two groups regarding the academic training of parents; thus, parents of psychiatric patients tended to have a lower level of schooling than parents in the somatic group (*p* = 0.048, Likelihood Ratio = 12,688, df = 6, Phi = 0.321 (moderate association power) (Figure 1)).

For each subject, the score for the involvement/friendship field was calculated; the higher the score, the higher the degree of involvement/friendship. The subjects in the somatic group had scores between a minimum of 10 and a maximum of 30, with an average +/− SD = 24.88 +/− 4.78 and a median of 26.5. The subjects in the psychiatric group had scores between a minimum of 7 and a maximum of 30 with an average +/− SD = 19.98 +/− 5.66 and a median of 19.50. Kolmogorov−Smirnov normality tests showed that the values in both groups had uneven distributions. The Mann−Whitney test showed that the median values for the involvement/friendship field in the somatic group were statistically significantly higher compared with those in the psychiatric group (U = 616.00, *p* = 0.00) (Figure 2).

For each subject, the score for the empathy/care/worry field was calculated; the higher the score, the higher the level of empathy/care/worry. The subjects in the somatic group had values of this score between a minimum of 11 and a maximum of 25, with an average +/− SD = 22.62 +/− 2.95 and a median of 23.5. The subjects in the psychiatric group had values between a minimum of 10 and a maximum of 25, with an average +/− SD = 20.35 +/− 4.74 and a median of 22. Kolmogorov−Smirnov normality tests showed a non-parametric distribution for the values of this score in both groups. The Mann−Whitney test showed that the median empathy/care/concern score values of the subjects in the somatic group were statistically significantly higher compared with those in the psychiatric group (U = 616.00, *p* = 0.041) (Figure 3).

For each subject, the score for the education/learning field was calculated. The subjects in the somatic group had values of this score between a minimum of 4 and a maximum of 20, with an average +/− SD = 16.04 +/− 3.77 and a median of 17. The subjects in the psychiatric group had values of this score between a minimum of 5 and a maximum of 20, with an average +/− SD of 13.78 +/− 3.93 and a median of 13. Kolmogorov−Smirnov tests showed that the values for this field followed non-parametric distributions in both groups. The Mann−Whitney test showed that the median values in the somatic group were statistically significantly higher compared with that of the psychiatric group values (U = 816.50, *p* = 0.03) (Figure 4).

For the conflict/aggression, avoidance, and rivalry domains, the t-test for independent samples showed that there were no statistically significant differences between the median values between the two groups.

In the somatic group, 66% of the patients had at least one sibling with an age greater than or equal to 11 years and 52% had siblings with younger than or equal to 10 years old. In the psychiatric group, 78% of the patients had at least one sibling of more than or equal to 11 years of age and 42% had siblings less than or equal to 10 years of age. There were no significant differences between the groups (*p*= 0.405, df = 2, Chi-square = 1810).

The subjects in the psychiatric group, whose siblings were at least 11 years old, had rivalry score values ranging from a minimum of 7 to a maximum of 35, with an average +/− SD of 15.28 +/− 5.612 and a median of 14. For those with siblings under or equal to 10 years of age, values ranging from a minimum of 13 to a maximum of 25 were obtained, with an average +/− SD of 18.73 +/− 3952 and a median of 18. The distributions of values for both groups were non-parametric. The Mann−Whitney test showed that there was a statistically significant difference between the medians of the values in the two groups, so where there were siblings under the age of 10, rivalry scores tended to be higher (U = 83.50, *p* = 0.02) (Figure 5).

The subjects in the somatic group, whose siblings were at least 11 years old, had rivalry score values ranging from a minimum of 8 to a maximum of 30, with an average +/− SD of 16 +/− 5116 and a median of 15. For those with siblings under or equal to 10 years of age, values ranging from a minimum of 13 to a maximum of 28 were obtained, with an average +/− SD of 18.82 +/− 4.391 and a median of 19. The distributions of values for both groups were non-parametric. The Mann−Whitney test showed that there was a statistically significant difference between the medians of the values in the two groups, so where there were siblings under the age of 10, the rivalry scores tended to be higher (U = 128.50, *p* = 0.045) (Figure 6).

The Mann−Whitney test showed that there were no statistically significant differences between the medians of the values in the two groups regarding the influence of the sibling age on the involvement/friendship, empathy/care/worry, education/learning, conflict/aggression, and avoidance domains.

Regarding the characterization of brothers in relation to age (≤10 years, > 10 years) and pathology (psychiatric/somatic), there were no statistically significant differences in relation to the total score of involvement/friendship (Factorial Anova: *p* = 0.860, df = 1), although the group trend was that the score for involvement decreased the higher the age of the siblings and the existence of a psychiatric pathology (Figure 7). There were no statistically significant differences regarding the empathy/care/worry, education/learning, conflict/aggression, rivalry, and avoidance domains related to the age and disorder.

## 4. Discussion

The theme of the study was based on the statistics estimated by recent studies, according to which up to 30% of children suffer from a chronic disease [10], and the data published in 2016 by the OECD report that up to 17% of children live with a sibling with a chronic condition [11]. Most of the studies on this topic have evaluated the family structure and the relationships between siblings focusing on the impact of physical and mental health and on the educational path/cognitive development of the healthy sibling. Our study, however, evaluated the dynamics of the relationship between the healthy sibling and the one with a chronic disease, the existence of rival behaviors, and the intensity of these behaviors depending on the age of the healthy sibling.

The first result that should be discussed in this section is that the siblings from a family with a child with a chronic pediatric condition tended to be more involved, to have a higher degree of engagement for the medical condition of the brother/sister, and be more willing to share age-specific activities for a lasting friendship relation. The scores for the involvement/friendship field were lower for the group of families with a child with a chronic mental health disorder. Possible explanations for these results could be the inability of the family or siblings to understand the psychiatric condition [12], or behavioral management difficulties during agitation, aggression, or oppositional episodes manifested in the family environment [13]. In the cases of families with a child with autism, it is even harder for their brothers/sisters to co-opt them in age-specific game activities [14], as some of the symptoms of autism are the inability to understand social relationships and to initiate and engage in symbolic play, especially at a young age.

According to our results, the healthy sibling showed a higher degree of empathy and concern towards the child with chronic somatic conditions, compared with the healthy sibling of children with mental health disorders.

The results for education/learning field demonstrated that in families with a child with a chronic somatic illness, the healthy sibling was more committed to teaching the sibling with a chronic somatic disease new skills, help to adapt to new situations, and to spend more time together teaching them how to behave in certain contexts. These results are predictable considering that it is much harder to interact, set limits, and obtain involvement reciprocity from a child who is facing, for example, neurodevelopmental disorders [15]. This trend is not as well highlighted in families where there is a child with chronic mental health problems, which brings into discussion the importance of affectionate bond and positive siblings’ attachment in the emotional and behavioral development of the healthy sibling [16,17].

Negative sibling attachment experiences can generate unfavorable effects, developing internalizing or externalizing disorders. Negative interactions with their sibling, poor involvement in group activities, and the tendency to engage in conflicts can generate feelings of loneliness and depression, as well as delinquent behaviors or conduct disorders [18].

Sharpe’s meta-analysis from 2002 on data concerning the siblings of children with a chronic illness found a statistically significant and negative overall effect for having a sibling with a chronic illness. The authors showed that psychological functioning, peer activities, and cognitive development scores were lower for siblings of children with a chronic illness compared with the healthy controls [19].

Considering the results of this analysis, the recommendation for professionals who treat children with chronic illnesses is to also pay attention to their siblings arises, as they could be at risk for experiencing negative psychological effects. Interventions such as psychoeducation sessions and support groups have been shown to enhance children’s psychological wellbeing, their awareness about disabilities, and their understanding of the family situation [20]. The effectiveness of these interventions for the assistance of the siblings of children with a chronic illness, especially a mental health condition, could be a focus of future research.

According to Gass et. al, there are positive benefits of living with a disabled sibling, such as greater compassion and emphatic initiative [21]. However, the results of our study showed a decreasing tendency for fraternal relationship involvement with advancing age, depending on the existence of a chronic mental disability.

Both in the somatic group and in the psychiatric group, the tendency to rivalry was higher for siblings under the age of 10. In relation to age (≤10 years, > 10 years) and pathology (psychiatric/somatic), there were no statistically significant differences in relation to the total scores of the other domains that we evaluated. In this regard, it is expected that the behavior of the healthy sibling towards the affected one, whether psychiatric or somatic, will have different characteristics, depending on the age and the moral judgment development level of the former. It is worth mentioning that moral judgment develops independently of cognition. If by the age of 10, the child can refrain from certain behaviors without challenging their family’s rules; starting at this age the child begins to challenge the rules that they no longer regard as immutable [22,23].

Even though moral reasoning develops over time, we have to take into account that these developmental stages in the healthy siblings are dependent of the social stimulation and familial contexts. At the age of 10, the child prioritizes their own needs over the needs of others. Sibling relationship quality has been shown to fluctuate throughout childhood and adolescence [24]. An explanation for these results could be the transformation of young-age rivalry into symptoms of negative affect (such as shame or embarrassment) in adolescence [24,25].

From the Piaget’s Theory publishment to the present literature, the understanding of moral physiology has been enriched with a new aspect, with moral judgement no longer being considered only the ability to choose between wright and wrong under favorable conditions [26]. In the last few years, psychological theories have presented a substantial diversity in terms of moral judgement, classifying this concept into four theoretical categories [27,28]:Blame judgements: reproach/blameWrongness judgement: morally wrong/immoralNorm judgement: permissible/obligatory/forbiddenEvaluations: bad/negative.

Previous studies have identified discrepancies between parent and sibling reports regarding the quality of intrafamilial relationships [29,30]. Thus, one of the limitations of our study is that it only analyzed the parental perspective. Studies on parent reports indicate negative aspects of their quality of life, because of their stress and the child’s diagnosis burden. Additionally, they indicate that the healthy siblings have a tendency to under report the difficulties they face in relation to their affected sibling, in order to formulate the answer they think is expected from them [31,32,33]. Another important limitation was that the order of birth of the child with a disability was not considered, as it is known that for a sibling relationship it is important whether a given child is the youngest or the oldest among siblings.

Another limitation of our study was correlated with the number of participants, which did not allow for the generalization or analysis of variables such as age, gender, or type of mental/somatic disorder. However, the results brought visibility and a comparative perspective related to the two groups, namely, chronic psychiatric and somatic diagnosis. Because of the small sample size, the SIB questionnaire could not be validated prior to the beginning of the study.

## 5. Conclusions

Our results highlight that the sibling relationship is a significant factor when identifying children at risk, and family-based intervention programs should be developed. These current findings provide a better perspective on the need for support programs in the case of children who face difficulties understanding and approaching a sibling with a chronic psychiatric disorder.

Living with a sibling with a chronic mental illness has a negative impact on their own psychological functioning, on long lasting peer activities, and may qualify as a risk factor for developing further psychopathology. The subject needs further scientific evaluation to identify current gaps. In order to provide directions for future research, we consider it of great importance that the development of new quantification methods for the severity of the disease and the degree of impairment of daily functionality in the affected child occurs, in order to better assess the impact of these factors on the psychological state of healthy siblings. Another future research direction could be the analysis of sibling relationships taking into account the gender/the order of birth of the child with a disability, and the comparative results of the parents.

## Figures and Tables

**Figure 1 children-10-00587-f001:**
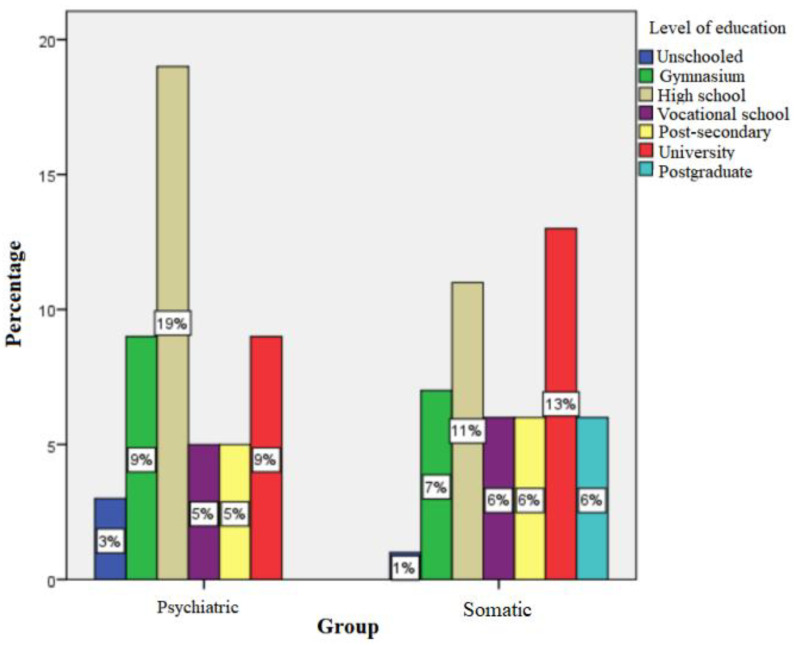
Level of school education of parents.

**Figure 2 children-10-00587-f002:**
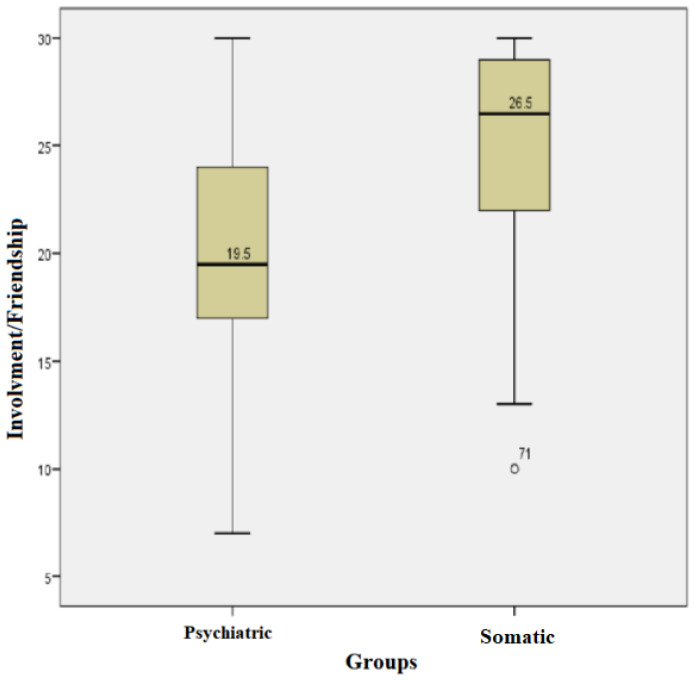
Involvement/friendship field for both groups.

**Figure 3 children-10-00587-f003:**
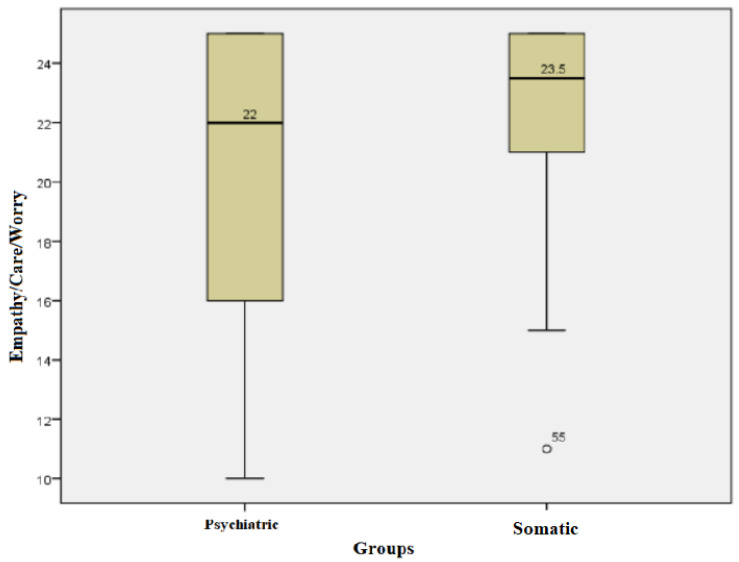
Empathy/care/worry field for both groups.

**Figure 4 children-10-00587-f004:**
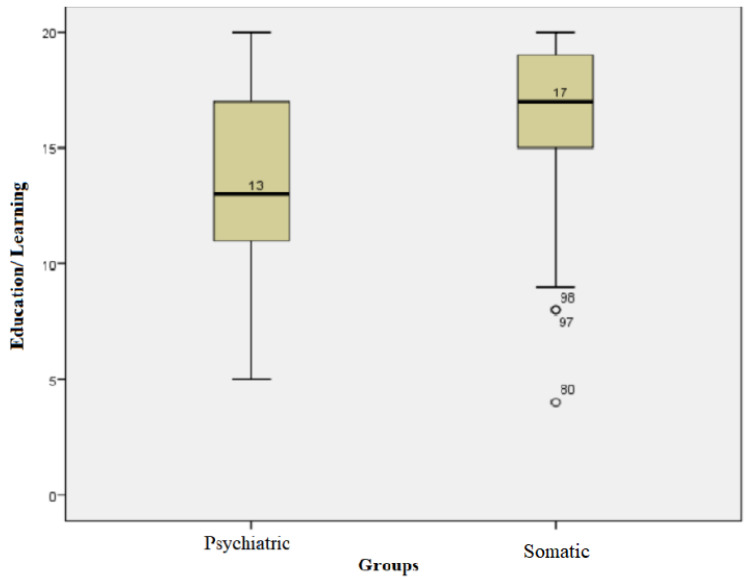
Education/learning field for both groups.

**Figure 5 children-10-00587-f005:**
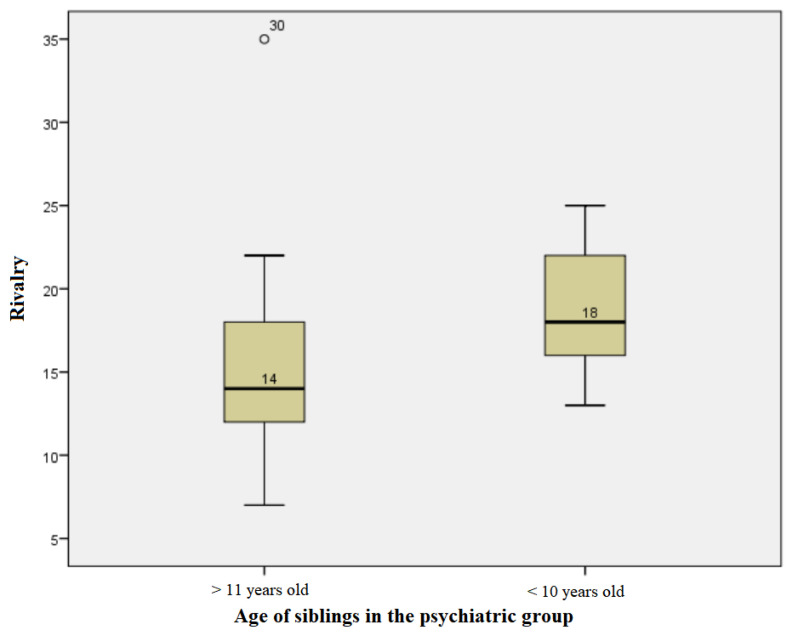
Age influence on rivalry scores in the psychiatric group.

**Figure 6 children-10-00587-f006:**
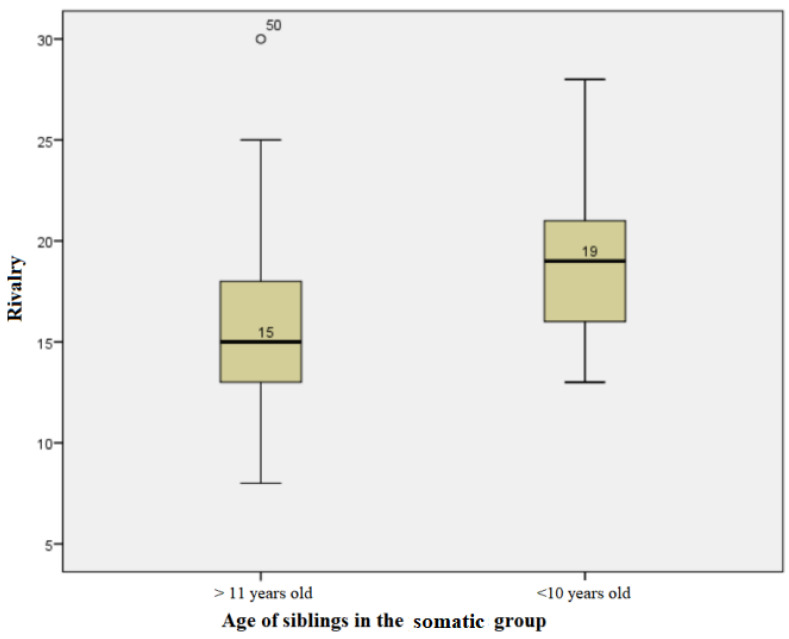
Age influence on rivalry scores in the somatic group.

**Figure 7 children-10-00587-f007:**
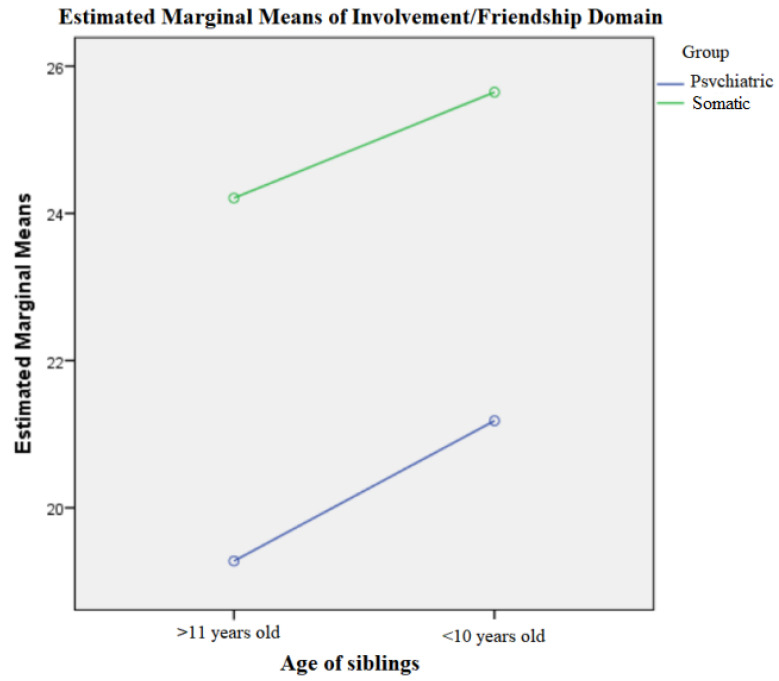
Involvement/friendship field in relation to age and disorder.

## Data Availability

Requests for the study data can be addressed to the authors. The raw data are not publicly available due to privacy considerations.

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
