# Peer review of "Sibling Relationship Dynamics in Families with a Child Diagnosed with a Chronic Mental Disorder versus a Somatic Condition"

_children, 2023, doi:10.3390/children10030587_

Round 1
Reviewer 1 Report
The research refers to an important issue, which is the relationship between siblings in families with a chronically ill child. The authors are particularly interested in whether the type of chronic disease (somatic vs. mental) may affect these relationships. For this purpose, they conducted research in a group of 50 families with a chronically somatically ill child and in a group of 50 families with a chronically mentally ill child. The subjects were parents who had completed the Sibling Inventory of Behavior (SIB) and a demographic questionnaire.
Alas, the reviewed text contains many ambiguities and errors that require correction.
1. On page 2 (line 85-86) the authors state that ‘Both parents gave their consent to participate in the study and both of them filled out the questionnaires’. However, in the 'Results' section, the authors do not explain whether the analyzed results are the average obtained from the result of the father and mother or in some other way. This issue requires clarification. By the way, comparing the results of fathers and mothers could lead to interesting conclusions.
2. On p. 2 (lines 88-89) the authors state that 'Only the attitude of brothers aged 7-10 years and 11-18 years was evaluated'. This statement raises serious doubts. First, the SIB is not a technique that is used to study children's attitudes, but their behavior. In this case, parents were asked to rate the occurrence of specific behaviors in their children and their frequency. Such wording may mislead the potential reader of this article. Secondly, if the behaviors of girls were indeed omitted, perhaps it should be indicated in the title that it is about sibling relations among boys. In addition, it should be mentioned in the ‘Limitations’ section.
3. The analysis of the results does not take into account the order of birth of a child with a disability, which is another limitation of this study. It is known that for a sibling relationship it is important whether a given child is the youngest or the oldest among siblings. This should also be mentioned in the 'Limitations' section.
4. Serious doubts are also related to the use of terms defining the studied groups. In the introductory part, the authors announce that they will compare sibling relationships in families with a somatically ill child vs. mentally ill. In this context, the term 'pediatric group' refers to both compared groups, as we are dealing with children here. Therefore, for the average reader it may be incomprehensible why it is used only in reference to families of children with somatic diseases. So I propose to replace the term ‘pediatric group’ with the term 'somatic group' throughout the text.
5. In the text of the article there are many incorrect and unclear expressions that may make it difficult for the average reader to understand. E.g. 'a child from fratria' (page 2, line 75), 'whiteout' (page 2, line 80), 'the characterization of fratria' (page 9, line 200), 'anterior research' (page 11, line 278) and so on. Therefore, I request that the entire text be corrected by a 'native speaker'.
In my opinion, the reviewed article in its current form does not qualify for publication, as it requires significant correction.
Author Response
Dear Reviewer,
We want to extend our thanks and appreciation for your time and effort in writing your comments that helped improve the manuscript. We trust that all of your comments have been addressed accordingly in this response letter and the revised manuscript.
The responses to your comments are found in the attached document.
Sincerely,

Reviewer 2 Report
Thank you for the opportunity to read this very interesting paper. The place of siblings in a family dynamic such as you describe is quite important. I have added a few comments on the paper - it is attached and the comments can be seen by using adobe. There are a few minor areas for revision / clarification.
In particular, the paper's place in the academic and clinical community might be enhanced by giving us some guideposts on where should the next steps in research focus.

Author Response

(The authors gave the same response as above.)

Reviewer 3 Report
I am attaching the pdf with some revision suggestions. Regarding the method, the evaluation of a statistician is needed.
In my opinion, with minor revisions the work can be published.

Author Response

(The authors gave the same response as above.)
